# Driving Effects and Spatial-Temporal Variations in Economic Losses Due to Flood Disasters in China

**Zhixiong Zhang [1,2], Qing Li [1,2], Changjun Liu [1,2,*] , Liuqian Ding [1,2], Qiang Ma [1,2] and Yao Chen [1,2]**

1   China Institute of Water Resources and Hydropower Research, Beijing 100038, China;
    zhixiongzhang1991@foxmail.com (Z.Z.); liqing@iwhr.com (Q.L.); dinglq@iwhr.com (L.D.);
    maqiang@iwhr.com (Q.M.); chenyao88520@126.com (Y.C.)
2   Research Center on Flood & Drought Disaster Reduction of the Ministry of Water Resources,
    Beijing 100038, China
*   Correspondence: lcj2005@iwhr.com; Tel.: +86-10-6878-1214

**Abstract:** The economic loss caused by frequent flood disasters poses a great threat to China's economic prosperity. This study analyzes the driving factors of flood-related economic losses in China. We used the extended Kaya identity to establish a factor decomposition model and the logarithmic mean Divisia index decomposition method to identify five flood-related driving effects for economic loss: demographic effect, economic effect, flash flood disaster control effect, capital efficiency effect, and loss-rainfall effect. Among these factors, the flash flood disaster control effect most obviously reduced flood-related economic losses. Considering the weak foundation of flash flood disaster prevention and control in China, non-engineering measures for flash flood prevention and control have been implemented since 2010, achieving remarkable results. Influenced by these measures, the loss-rainfall effect also showed reduction output characteristics. The demographic, economic, and capital efficiency effects showed incremental effect characteristics. China's current economic growth leads to an increase in flood control pressure, thus explaining the incremental effect of the economic effect. This study discusses the relationship between flood-related economic loss and flash flood disaster prevention and control in China, adding value for the adjustment and formulation of future flood disaster prevention policies.

**Keywords:** economic losses from flood disasters; flash flood disaster control; Kaya identity; LMDI technique decomposition method

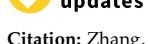



## 1. Introduction

Flooding has the highest frequency of all natural disasters worldwide. From 2000 to 2019, flooding accounted for 44% of the total number of natural disasters [1]. About two thirds of China's land area regularly face the threat of floods of different types and degrees of danger [2]. Moreover, the economic losses caused by these floods have seriously hindered the sustainable development of China's economic society [3]. Therefore, investigating the main factors affecting the economic losses related to flood disasters has important reference significance for flood disaster prevention and control policymaking in China.

Considering the economic losses caused by flood disasters, many researchers have performed in-depth analyses and research, obtaining corresponding research results. Among them, Jiang performed a comprehensive analysis of the characteristics of flood disaster losses in China from 1950 to 2016 and proposed that flood-related economic losses in China showed a downward trend [4]. Jiang mainly defines the concept and limitations of the indirect economic losses related to urban flood disasters, highlighting that the correlation between industrial loss and resources is a substantial part of the indirect economic loss caused by urban flood disasters [5]. Some researchers introduced the dynamic computable general equilibrium model for comprehensive disaster-related economic loss assessment and constructed an equilibrium model of rainstorm and flood disasters [6–8]. Their results

show that the occurrence of rainstorms and floods will affect social and economic development in the current year, as well as have a significant impact on economic development in later periods [6–8]. These studies mainly focus on calculating the direct and indirect economic losses related to flood disasters. However, in China, 70% of flood-related economic losses are due to flash flood disasters [9]; nevertheless, research on the relationship between the economic losses due to flood disasters and flash flood disasters is relatively rare.

Considering the weak foundation of flash flood disaster prevention and control in China [10], the Ministry of Water Resources officially started implementing the special construction of flash flood disaster prevention and control infrastructure in 2010 [11]. During the first stage of the construction for flash flood disaster prevention and control, from 2010 to 2012, a total of 11.7 billion Yuan was invested in the construction of a non-engineering system for flash flood disaster prevention and control in 2058 counties of 29 provincial-level administrative regions. During the second stage, from 2013 to 2015, a total of 14.3 billion Yuan was invested nationwide to investigate and evaluate flash flood disasters, construct non-engineering measures for the prevention and control of flash floods, and to prevent flash flood gulley erosion. A total of 9.2 billion Yuan was invested in the third phase of the program, which lasted from 2016 to 2020. The main focus of this stage was optimizing and improving the non-engineering measures for the prevention and control of flash flood disasters, utilizing the results of flash flood disaster investigations and evaluations, and continuing to carry out the construction of disaster mass monitoring and mass preventing of flash floods [12,13].

Based on the above, this study will fully consider the dual nature and social attributes of the disaster-causing factors of floods, by employing the Kaya identity and a logarithmic mean Divisia index (LMDI) approach to quantitatively measure the driving effects of inter-annual changes in economic loss related to flood disasters in China. The Kaya identity is used to decompose the driving factors and the LMDI method is used to determine the size of each influencing factor. We also explore the influence of the flash flood disaster prevention investment, economy, annual rainfall, and other factors on the changes in flood-related economic loss.

## 2. Methods

### 2.1. Kaya Identity

The Kaya identity was proposed by Japanese scholar Yoichi Kaya in 1989 [14] and was originally used for carbon emissions research [15,16]. After years of research expansion, it is currently widely used in the field of energy research. The Kaya identity uses a simple mathematical formula to explain the relationship between the macro overall social and economic factors and describes these using simple mathematical relationships that take into account the national level of carbon emissions associated with human production and living in four elements. As it employs a simple mathematical formula to change drivers, the Kaya identity has the advantage of strong explanatory power and is thus widely used across different fields.

The Kaya identity decomposes carbon emissions into four influencing factors, and the expression formula is as follows:

$$C = P \times \left(\frac{G}{P}\right) \times \left(\frac{E}{G}\right) \times \left(\frac{C}{E}\right) \tag{1}$$

where $G$ = gross domestic product (GDP); $E$ = energy consumption; $G/P$ = per capita GDP; $E/G$ = energy intensity; and $C/E$ = carbon intensity in energy consumption.

The advantage of the Kaya identity, compared with other models for studying driving factors of carbon emissions, lies in the fact that researchers can expand the Kaya identity according to their own research needs and add other influencing factors to study its influence on the change in research objects. Recently, many researchers have applied this method to different research fields [15,16].

To better describe the effects of different factors on flood-related economic losses, we use the Kaya identity to expand the decomposition of flood-related economic loss.

We define the variable $C$ as the economic losses from flood disasters in province i. Specifically, the economic loss due to floods in each province is decomposed into a product of five factors: the resident population or the demographic effect of province i ($P$); the per capita GDP or the economic effect of province i ($G/P$), representing current operational economic status; the flash flood disaster prevention and control of investment–GDP ratio or the flash flood disaster control effect of province i ($W/G$), representing the investment intensity in disaster prevention; the rainfall–flash flood disaster prevention and control of investment ratio or the capital efficiency effect of province i ($R/W$), representing the intensity of rainfall faced per unit defense fund; and the economic losses of flood disasters–rainfall ratio or the loss-rainfall effect of province i ($C/R$). Then, we denote the five factors $p$i, $g$i, $w$i, $e$i, and $a$i, as shown in the following expression.

$$C = P \times \left(\frac{G}{P}\right) \times \left(\frac{W}{G}\right) \times \left(\frac{R}{W}\right) \times \left(\frac{C}{R}\right) = p \cdot g \cdot w \cdot e \cdot a \tag{2}$$

*2.2. Logarithmic Mean Divisia Index*

The LMDI was first proposed by Professor Ang from Singapore [17–19]. This method was at first applied in carbon emissions research, mainly to analyze energy intensity change. The LMDI is a common factor decomposition model in global research and gives perfect decomposition [20]; this means that the results do not contain an unexplained residual term, which simplifies the result interpretation.

According to the LMDI decomposition model, the total effect $\Delta C$ on the change value of flood-related economic loss in the base period and the year t is called the total effect, representing demographic effect ($P_{effect}$), economic effect ($g_{effect}$), flash flood disaster control effect ($w_{effect}$), capital efficiency effect ($e_{effect}$), and loss-rainfall effect ($a_{effect}$). When the calculated effect value is positive, this index has an incremental effect on the direct flood-related economic loss and will lead to an increase in disaster loss. Conversely, when the effect value is negative, the index has a reduction effect on the direct flood-related economic loss, which can reduce the economic loss. The five driving factor relationships can be expressed as:

$$\Delta C = C_t - C_0 = p_{effect} + g_{effect} + w_{effect} + e_{effect} + a_{effect} \tag{3}$$

$$p_{effect} = \frac{C_t - C_0}{\ln C_t - \ln C_0} \ln \left(\frac{p_t}{p_0}\right)$$

$$g_{effect} = \frac{C_t - C_0}{\ln C_t - \ln C_0} \ln \left(\frac{g_t}{g_0}\right)$$

$$w_{effect} = \frac{C_t - C_0}{\ln C_t - \ln C_0} \ln \left(\frac{w_t}{w_0}\right)$$

$$e_{effect} = \frac{C_t - C_0}{\ln C_t - \ln C_0} \ln \left(\frac{e_t}{e_0}\right)$$

$$a_{effect} = \frac{C_t - C_0}{\ln C_t - \ln C_0} \ln \left(\frac{a_t}{a_0}\right)$$

where $C_0$ = flood-related economic loss in the base period; $C_t$ = flood-related economic loss in the year $t$.

*2.3. Study Object and Data Sources*

In this study, various sources were used to collect the data of 29 provincial areas in China, from 2010 to 2020. These areas included 21 provinces, 3 province-level megacities (Beijing, Tianjin, and Chongqing), and 5 autonomous regions. Due to the lack of relevant statistical data, Taiwan and two special administrative regions—Hong Kong and

Macao—were excluded from the study. Furthermore, as Shanghai and Jiangsu Province are not included in the flash flood disaster prevention and control project area, to ensure the consistency of data integrity of all provinces and cities, Shanghai and Jiangsu were excluded from this study.

The key data sources were the China Statistical Yearbook 2010–2020, obtained from the website of the National Bureau of Statistics of the People's Republic of China, and the China Water Resources Bulletin and the Bulletin on Flood and Drought Disasters in China 2010–2020, obtained from the website of the Ministry of Water Resources of the People's Republic of China.

The list of counties involved in flash flood disaster prevention and control was obtained from the National Flash Flood Prevention and Control Planning. The funding data of the prevention and control project were derived from the implementation plans of national flash flood disaster prevention and control projects in China. Flash flood disaster prevention and control projects are implemented by the county as a unit, the main task of which is to establish flash flood disaster monitoring and an early warning platform for flash flood disasters at the county level. This is accomplished through real-time dynamic monitoring of the water and rain situation to achieve flash flood disaster prevention and control work. In addition, as the funds for flash flood disaster prevention and control projects were distributed in stages, the public data on the projects' construction funds in each county were not available. Therefore, this study adopts the method of average allocation relating to the data of flash flood disaster prevention and control funds.

## 3. Results and Discussion

### 3.1. Temporal and Spatial Distribution of Flood-Related Economic Losses in China

According to the statistical data from the China Flood and Drought Disaster Prevention Bulletin, the average annual flood disaster loss in China from 2010 to 2020 was 237.326 billion Yuan (Figure 1). The flood-related economic loss in 2010 was the highest in the study period, at 374.543 billion Yuan, while the loss in 2011 was the lowest, at 130.127 billion Yuan. In terms of the spatial distribution of flood-related economic losses, flood disasters in China show a trend of being high in the south and low in the north (Figure 2). From 2010 to 2020, Sichuan, Guangdong, Hunan, Zhejiang, and Jiangxi were the top five provinces, highest to lowest, in terms of annual average economic losses from floods [21]. These five provinces are in the southern part of China, where rainfall is abundant and rivers are widespread, making these areas prone to flash floods. Sichuan Province had the highest average annual economic loss, at 22.130 billion Yuan. According to the statistical data and existing research [22,23], flood-related economic losses and the frequency of floods have both shown an increasing trend in Sichuan Province since 2014. Moreover, the frequency of flash flood disasters in Sichuan Province accounted for about 30% of the total number of floods in China [24]. Ningxia, Tianjin, Qinghai, Xizang, and Xinjiang had lower flood-related economic losses. These provinces are in the northern part of China, which has low rainfall and an arid climate, and are not prone to flash floods.

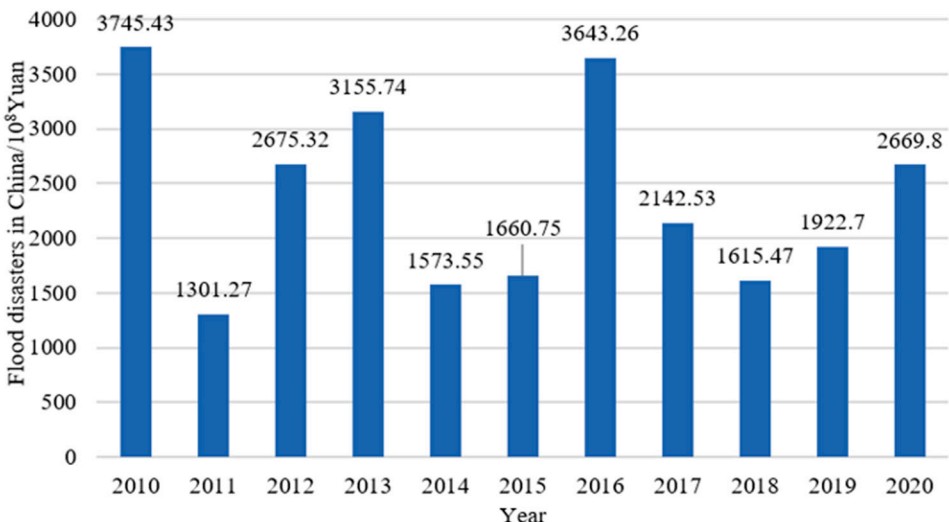

**Figure 1.** China's flood-related economic losses from 2010 to 2020. Data from the China Flood and Drought Disaster Prevention Bulletin, 2010–2020.

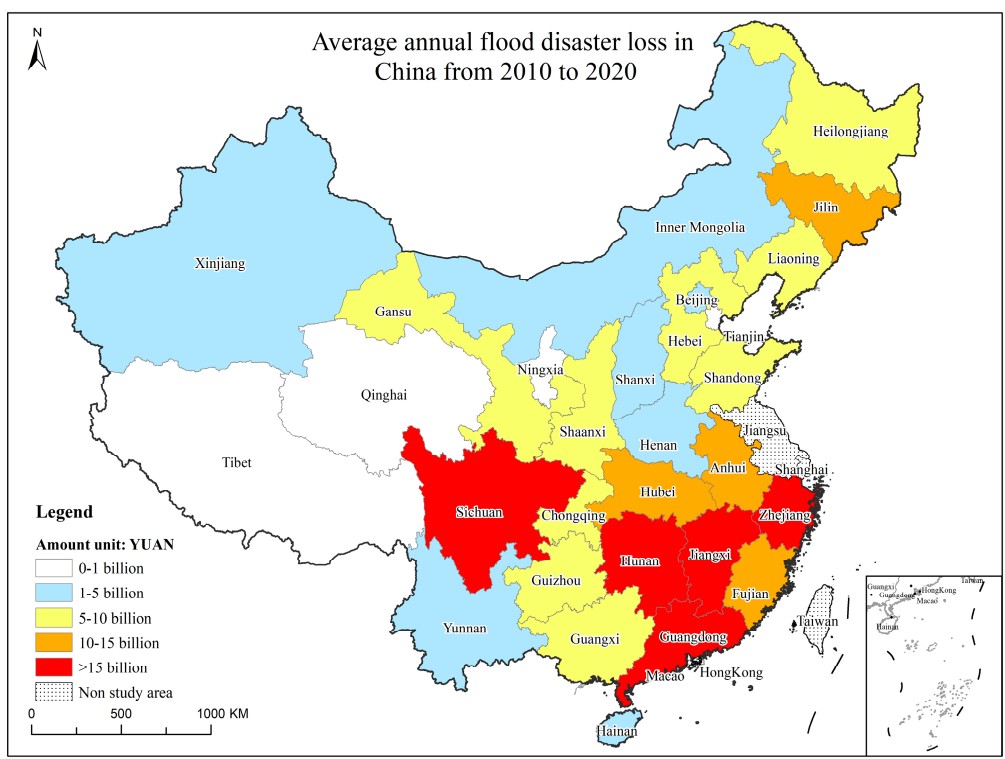

**Figure 2.** China's flood-related economic losses from 2010 to 2020.

### 3.2. Analysis of the Driving Factors of Flood-Related Economic Loss

Based on the LMDI decomposition model, this study analyzed the main driving factors affecting the direct economic losses related to flood disasters in China from 2010 to 2020. The demographic effect, economic effect, flash flood disaster control effect, disaster prevention pressure effect, and loss-rainfall effect on flood-related economic loss in 29 provinces of China were quantitatively analyzed (Tables 1 and 2). The results clearly show the reduction effect of the implemented flash flood disaster control measures. As the construction of flash flood disaster prevention and control continue, the amount of money invested in the flash flood disaster prevention and control work to reduce the economic loss plays an important role. The economic effect was characterized by obvious incremental effect. China is in

the later stage of industrialization and the middle stage of urbanization [25]. Therefore, investment in urban infrastructure construction is increasing greatly [26], and the growth of regional economic density will also bring greater challenges in flood control work. It is particularly necessary to focus on the frequent occurrence of urban waterlogging disasters in recent years, which has become a common problem in major cities [27]. The demographic effect was characterized by an incremental effect; however, the effect value is relatively low, compared with the economic effect, which is caused by the low population growth rate in China. The loss-rainfall effect was characterized by a reduction effect; this effect reflects the relationship between the flood-related economic loss and the annual rainfall intensity, indicating that disaster prevention ability is strengthened under the same rainfall intensity. On the whole, the capital efficiency effect showed an incremental effect, indicating that there is still at present great pressure on capital investment to cope with the prevention and control work of heavy rainfall.

**Table 1.** Effect decomposition of China's flood-related economic losses (in Yuan).

| Year | Loss-Rainfall Effect | Capital Efficiency Effect | Flash Flood Disaster Control Effect | Economic Effect | Demographic Effect | Total |
|---|---|---|---|---|---|---|
| 2010–2011 | −2033.789 | −410.3707 | −378.5802 | 364.3994 | 14.1807 | −2444.16 |
| 2011–2012 | 1056.044 | 318.0058 | −200.9019 | 186.739 | 14.1629 | 1374.05 |
| 2012–2013 | 659.6843 | −761.9825 | 320.2077 | 245.3544 | 17.1559 | 480.4199 |
| 2013–2014 | −1494.108 | −88.0821 | −190.0587 | 174.8112 | 15.2475 | −1582.19 |
| 2014–2015 | −9.8524 | 95.3566 | −117.6909 | 111.419 | 7.9674 | 87.1997 |
| 2015–2016 | 1731.183 | 2655.194 | −2611.375 | 191.0343 | 16.4746 | 1982.51 |
| 2016–2017 | −1236.258 | −264.4714 | −302.4272 | 286.6552 | 15.7719 | −1500.73 |
| 2017–2018 | −576.1079 | 49.0477 | −156.3971 | 149.3445 | 7.0526 | −527.0602 |
| 2018–2019 | 389.8009 | −82.5708 | −171.6208 | 165.767 | 5.8539 | 307.23 |
| 2019–2020 | 561.9533 | 185.1468 | −46.1579 | 42.8678 | 3.2902 | 747.1002 |
| Effect average | −95.145 | 169.5273 | −385.5002 | 191.8392 | 11.7158 | |
| Effect standard deviation | 1208.93 | 927.548 | 803.9399 | 90.1191 | 5.1031 | |
| Effect coefficient of variation | −12.7062 | 5.4714 | −2.0854 | 0.4698 | 0.4356 | |

**Table 2.** Effect decomposition of flood-related economic losses in China (in Yuan).

| Region | Loss-Rainfall Effect | Capital Efficiency Effect | Flash Flood Disaster Control Effect | Economic Effect | Demographic Effect | Total |
|---|---|---|---|---|---|---|
| Beijing | 0.3007 | −0.4130 | −1.3098 | 1.1466 | 0.2754 | 0.0000 |
| Tianjin | 0.6224 | −0.7810 | −0.0749 | 0.1827 | 0.0449 | −0.0059 |
| Hebei | 0.0623 | 6.3323 | −10.7380 | 3.4588 | 0.2147 | −0.6699 |
| Shanxi | −1.4803 | 3.3480 | −3.7116 | 1.3850 | −0.0441 | −0.5030 |
| Inner Mongolia | −3.2358 | 2.0408 | −1.3984 | 2.0413 | −0.0950 | −0.6471 |
| Liaoning | −20.0179 | −8.3127 | −2.5122 | 4.9932 | −0.1004 | −25.9500 |
| Jilin | −43.0281 | −5.9829 | −3.0722 | 2.9943 | −0.7361 | −49.8250 |
| Heilongjiang | −0.3310 | −0.3150 | 1.2548 | 0.9660 | −1.0277 | 0.5471 |
| Zhejiang | 3.1282 | 3.1035 | −20.9594 | 9.7150 | 2.6387 | −2.3740 |
| Anhui | 44.4181 | 25.9407 | −26.8928 | 8.0031 | 0.2319 | 51.7010 |
| Fujian | −17.0980 | 25.9043 | −38.1183 | 9.5228 | 1.1102 | −18.6790 |
| Jiangxi | −10.6887 | 1.7504 | −19.2507 | 12.2608 | 0.1662 | −15.7620 |
| Shandong | −3.7440 | −5.1765 | −0.8274 | 1.9150 | 0.3940 | −7.4389 |
| Henan | −13.1402 | 4.1320 | −5.9747 | 1.7968 | 0.1081 | −13.0780 |
| Hubei | 1.1752 | 29.3494 | −35.1599 | 10.3178 | 0.0956 | 5.7781 |
| Hunan | −6.0440 | 9.1161 | −25.6694 | 12.7677 | 0.1605 | −9.6691 |
| Guangdong | −5.1864 | 6.0111 | −27.7194 | 12.9568 | 3.4269 | −10.5110 |
| Guangxi | 3.2246 | 4.4470 | −8.9474 | 4.9755 | 0.6034 | 4.3031 |
| Hainan | −9.3221 | 0.9359 | −7.9598 | 4.0317 | 0.6672 | −11.6471 |

**Table 2.** *Cont.*

| Region | Loss-Rainfall Effect | Capital Efficiency Effect | Flash Flood Disaster Control Effect | Economic Effect | Demographic Effect | Total |
|---|---|---|---|---|---|---|
| Chongqing | 8.8715 | 6.8277 | −13.4133 | 7.2321 | 0.7579 | 10.2759 |
| Sichuan | −2.3962 | −0.9183 | −19.4274 | 19.5975 | 0.5814 | −2.5630 |
| Guizhou | 2.8052 | 8.5286 | −13.3576 | 5.7722 | 0.4985 | 4.2469 |
| Yunnan | 2.2289 | 5.0178 | −9.4595 | 4.7431 | 0.1117 | 2.6420 |
| Tibet | −0.5672 | 0.7986 | −1.6222 | 0.7790 | 0.1528 | −0.4590 |
| Shaanxi | −18.7190 | 1.4214 | −7.2012 | 6.6924 | 0.3425 | −17.4639 |
| Gansu | 3.8549 | −0.8067 | −3.1115 | 5.4300 | −0.1697 | 5.1970 |
| Qinghai | −0.3791 | 0.3224 | −0.6531 | 0.3436 | 0.0222 | −0.3440 |
| Ningxia | −0.1595 | 0.5062 | −0.7422 | 0.2460 | 0.0485 | −0.1010 |
| Xinjiang | −3.3441 | 2.9019 | −4.3370 | 1.1898 | 0.2564 | −3.3330 |
| National total | −95.1450 | 169.5273 | −385.5002 | 191.8392 | 11.7158 | −107.5629 |

To simplify and analyze the influence of the driving effects on the time series changes of flood-related economic losses more directly, we created a spatial clustering to illustrate the spatial variation in the driving effects, based on an ISODATA clustering model [28]. In this model, the clustering is specific to each type of drive effect analysis.

3.2.1. Demographic Effect

The demographic effect was the weakest in terms of change in flood-related economic loss, with an effect value of 1.17158 billion Yuan. As China enters a stage of low fertility [29], the population growth rate is relatively slow, with the population in some provinces showing a downward trend. This is the main reason for the low demographic effect value.

One region in which demographic effect was high is Guangdong (Figure 3). During the study period, the demographic effect value in the province was 342.69 million Yuan, showing the most obvious incremental effect among all provinces. According to statistical data, Guangdong is the most populous province in China, and during the study period, its population growth showed a steady upward trend, with an increase of 21.83 million people from 2010 to 2020. The increase in population also puts higher demands on the flood prevention work.

The regions with medium levels of demographic effect are Zhejiang, Fujian, and Chongqing (Figure 3). Zhejiang and Fujian are in the southeast coastal area of China, while Chongqing is in the southwest. These three regions have high population density, and all of them are flood prone provinces. Statistics show that the population of these three provinces are increasing and therefore the demographic effect was also incremental. However, compared with the high demographic effect area, these areas' effect values are lower.

The regions with low levels of demographic effect include the following 25 provinces: Beijing, Tianjin, Hebei, Shanxi, Inner Mongolia, Liaoning, Jilin, Heilongjiang, Anhui, Jiangxi, Shandong, Henan, Hubei, Hunan, Guangxi, Hainan, Sichuan, Guizhou, Yunnan, Tibet, Shaanxi, Gansu, Qinghai, Ningxia, and Xinjiang (Figure 3). Among them, the demographic effect values of Shanxi, Inner Mongolia, Liaoning, Jilin, Heilongjiang, and Gansu showed a decreasing effect, as the population number of these six provinces showed a decreasing trend during the study period; therefore, the population pressure in terms of flood disaster prevention also showed a decreasing trend. The other 19 provinces had average demographic effects of around 25.61 million Yuan. Although the characteristics of positive effect would lead to increased flood-related economic losses, overall, they account for a relatively small effect.

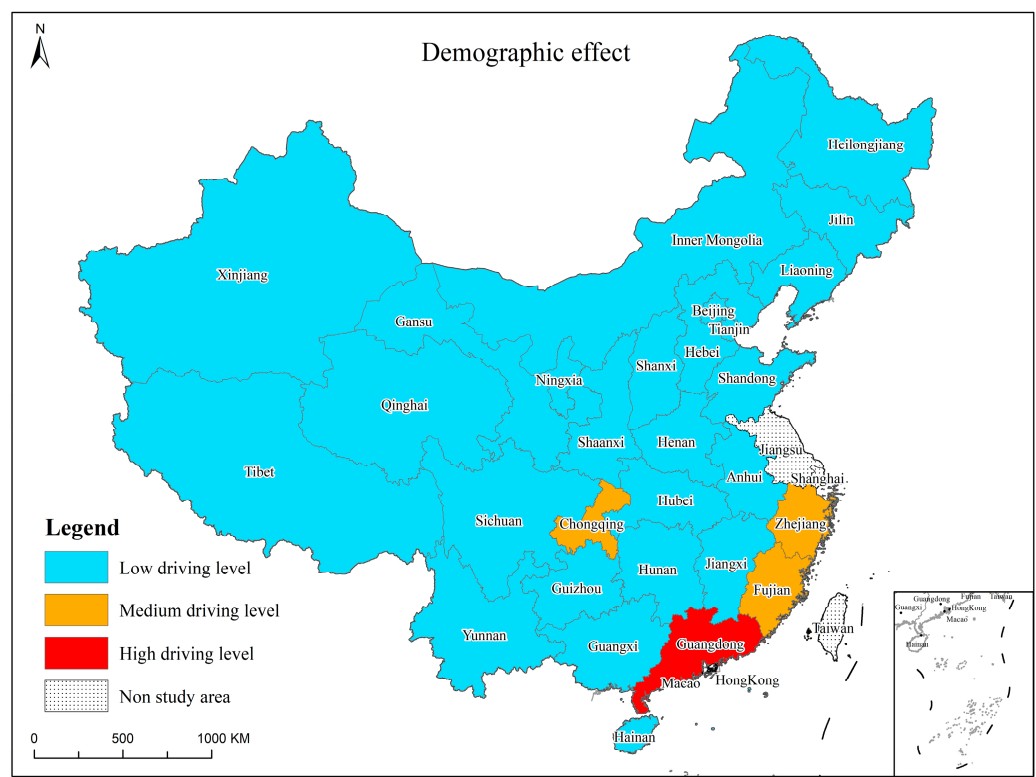

**Figure 3.** Demographic effect driving distribution.

### 3.2.2. Economic Effect

During the study period, the economic effect value of flood-related economic loss was 19.18392 billion Yuan, making it the most obvious incremental driving effect. From 2010 to 2020, China's national economy has maintained an average annual growth rate of more than 7% (Figure 4). The country's rapid economic growth also poses a growing challenge in terms of flood disaster prevention. The economic effects of flood-related economic loss in all provinces are also incremental.

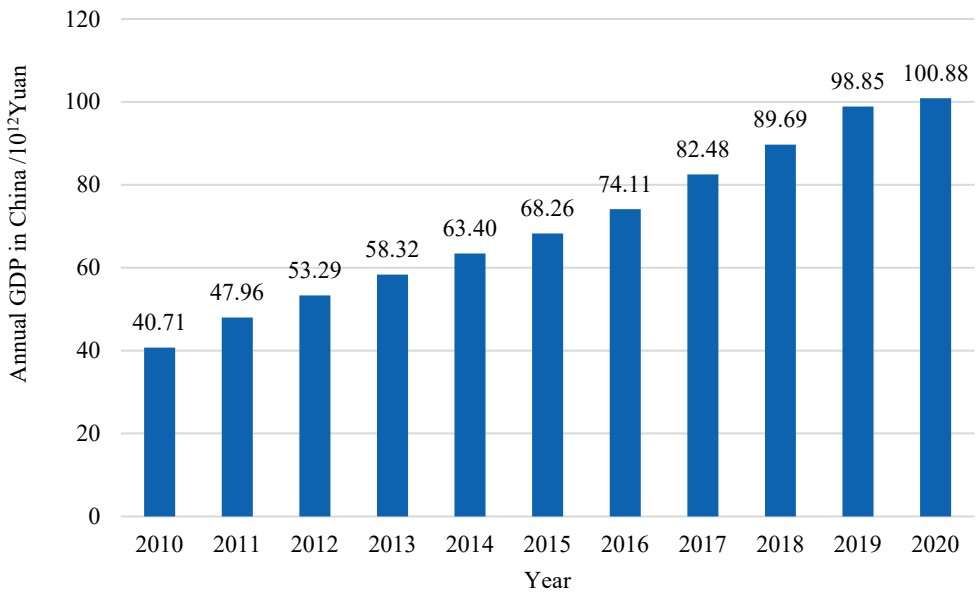

**Figure 4.** China's GDP from 2010 to 2020. Data from the China Statistical Yearbook 2010–2020.

Jiangxi and Sichuan showed the highest driving effect of flood-related economic losses, with the average effect of these two provinces during the study period as high as 1.45506 billion Yuan (Figure 5). The two provinces had similar GDPs per capita and are both prone to flash floods and flood disasters. During the study period, the average annual economic losses caused by flood disasters in Jiangxi and Sichuan ranked second and fifth, respectively, among the 29 provinces. With the continuous economic development, the per capita GDP in both Jiangxi and Sichuan has also increased significantly, which brings new challenges in terms of flood disaster prevention work; therefore, the economic effect was at a high driving level.

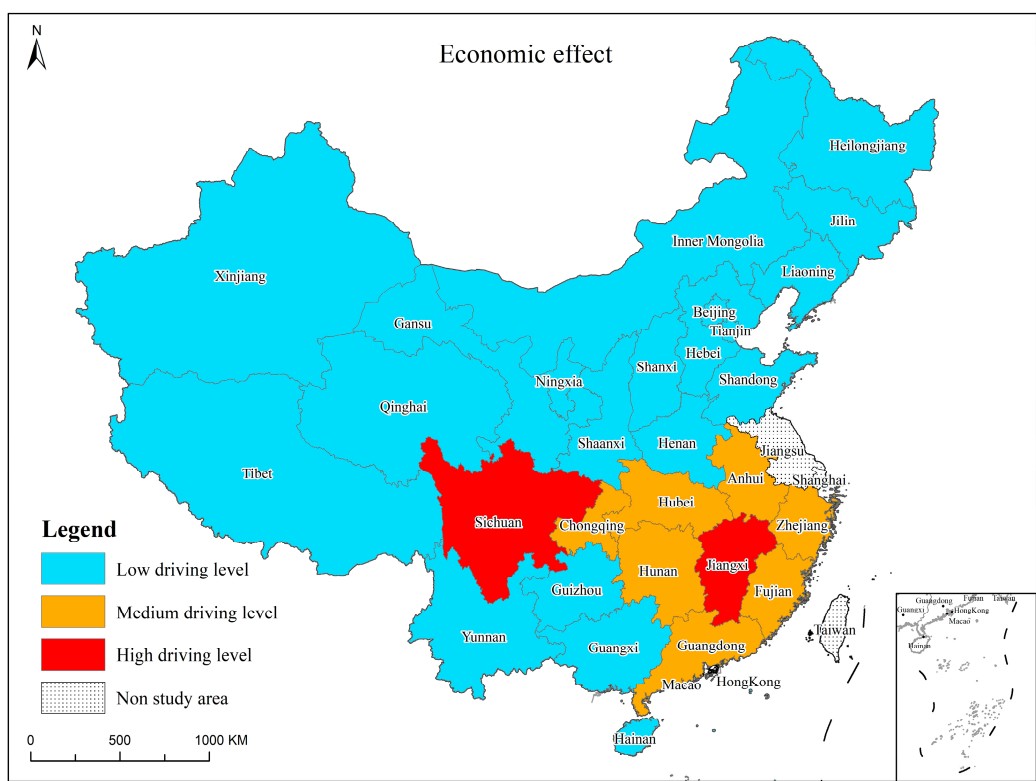

**Figure 5.** Economic effect driving distribution.

The provinces with medium driving levels of economic effect were Zhejiang, Anhui, Fujian, Hubei, Hunan, Guangdong, and Chongqing (Figure 5). According to statistical data, the economy of these seven provinces showed a general upward trend during the study period; moreover, the per capita GDPs of Zhejiang and Fujian ranked first and second in the country, with both provinces' per capita GDP exceeding 100,000 Yuan. In terms of spatial distribution, most of these provinces are in Southeast and South Central China, where annual rainfall is abundant. Therefore, these provinces are also prone to flood and flash flood disasters. Zhejiang, Fujian, and Guangdong are also typhoon-prone areas, and the flood prevention undertaking is particularly heavy in these provinces. The average annual economic losses caused by floods in these seven provinces were all among the top nine. Even though the continuous economic growth in these provinces will cause an increase in the difficulty of flood disaster prevention and control, the effect value was slightly lower than that of the high driving effect provinces.

There were 20 provinces that had low economic driving effects: Beijing, Tianjin, Hebei, Shanxi, Inner Mongolia, Liaoning, Jilin, Heilongjiang, Shandong, Henan, Guangxi, Hainan, Guizhou, Yunnan, Tibet, Shaanxi, Gansu, Qinghai, Ningxia, and Xinjiang (Figure 5). Among them, the per capita GDPs of Beijing and Tianjin in 2020 exceeded 100,000 Yuan, and the per capita GDPs of Inner Mongolia and Shandong exceeded 70,000 Yuan. These four provinces are in the northern region and their flood-related economic loss in recent years is

significantly lower than that of provinces and cities in the south. As their effect value is relatively small, they have a low economic driving effect. Most of the other 16 provinces are in Northern China, where economic losses from floods are relatively low; however, the per capita GDPs of these regions are significantly lower than those of regions with high and medium economic driving effects.

### 3.2.3. Flash Flood Disaster Control Effect

The flash flood disaster control effect showed an obvious reduction effect and had the highest absolute value among all five effects. This shows that the implementation of flash flood disaster prevention and control projects has greatly reduced flood-related economic losses in China. The continuous development of flash flood disaster prevention and control plays a positive role in reducing flood-related economic losses in China.

The provinces with high levels of flash flood disaster control effects were Shanxi, Anhui, Fujian, and Hubei (Figure 6). During the study period, the average effect of flash flood disaster prevention and control in these four provinces was −3.37905 billion Yuan, and the development of flash flood disaster work greatly reduced disaster-related loss in these regions. Shanxi, located in Northern China, is a loess-covered mountain plateau, with mountains and hills accounting for more than 80% of the province's total area [30]. Flash flood disaster prevention has always been a focus point and a difficulty in terms of Shanxi flood control. Anhui and Hubei are in the central and southern parts of China, while Fujian is in the southeast coastal area, an area that experiences more serious floods [21].

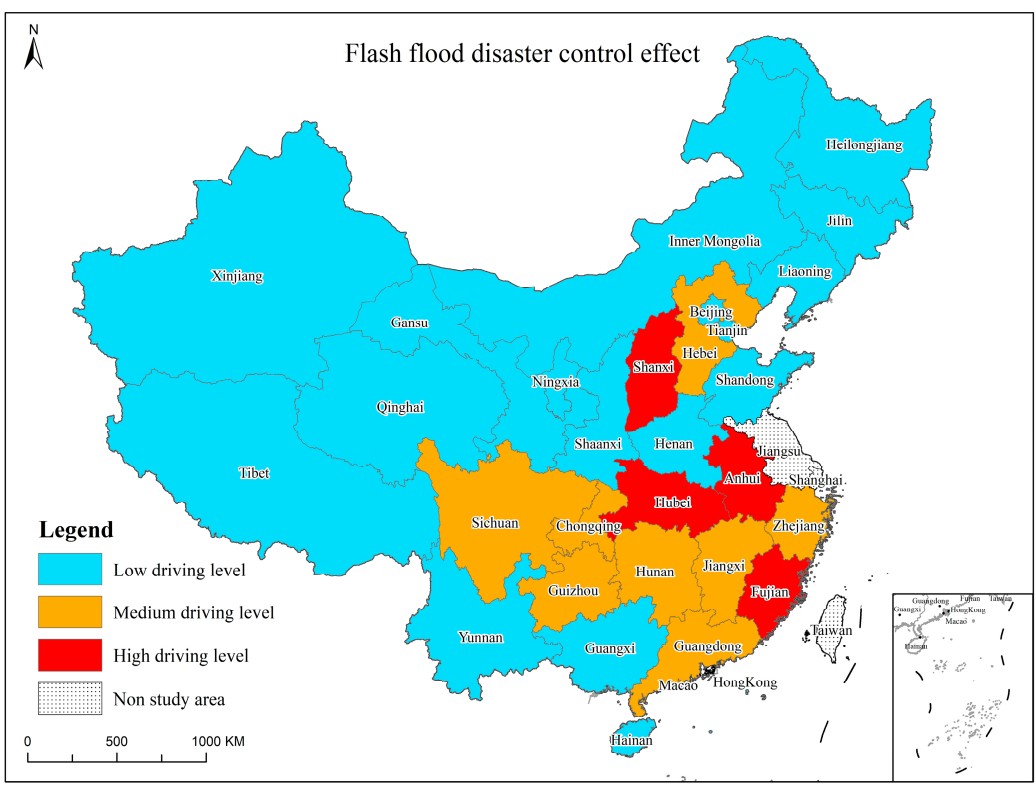

**Figure 6.** Flash flood disaster control effect driving distribution.

Provinces with a medium level of flash flood disaster control effect include Hebei, Zhejiang, Jiangxi, Hunan, Guangdong, Chongqing, Sichuan, and Guizhou (Figure 6); these provinces are mainly located in Southeast and Central China. The flash flood disaster prevention and control effect in these provinces has a reduction effect. Hebei is located at the eastern foot of Taihang Mountain, and mountainous areas account for 52.7% of the total area of the province [31]. Flash floods caused by local heavy rains occur frequently. Zhejiang, Jiangxi, Hunan, Guangdong, Chongqing, Sichuan, and Guizhou are in the south

of China, an area prone to flash flood disasters [21]. Through the flash flood disaster prevention and control projects, these provinces have established a sound flash flood disaster prevention system, which has played a significant role in disaster prevention and mitigation, reducing casualties, and improving the information level of grassroots water conservancy; it also had a positive impact on reducing flood-related economic losses [9–13].

The regions with low levels of flash flood disaster control effect include Beijing, Tianjin, Inner Mongolia, Liaoning, Jilin, Heilongjiang, Shandong, Henan, Guangxi, Hainan, Yunnan, Tibet, Shaanxi, Gansu, Ningxia, and Xinjiang (Figure 6). Heilongjiang was the only province with an incremental effect value of flash flood prevention; however, the effect value was small. Although Guangxi, Hainan, and Yunnan are in Southern China—an area prone to flash flood disasters—the average annual economic losses related to floods from 2010 to 2020 were relatively low; therefore, they are at a low driving level. Most of the other 12 provinces are in the northern part of China, and the flood-related economic loss was lower than that of the provinces with high or medium driving force.

### 3.2.4. Capital Efficiency Effect

The capital efficiency effect of flood-related economic loss reflects the relationship between regional flash flood disaster input and annual rainfall. The effect values of most provinces were incremental. Due to large variations in annual precipitation in each region and the relatively stable investment of flash flood disaster prevention funds, the inter-annual variation in the effect value of each province is also apparent.

The provinces with a high driving level of capital efficiency effect were Shanxi, Anhui, Fujian, and Hubei, and the capital efficiency effect was incremental (Figure 7). From 2010 to 2020, the average annual rainfall in Shanxi, Anhui, Hubei, and Fujian was 540 mm, 1272 mm, 1182 mm, and 1778 mm, respectively. Flash floods are the most critical type of flood disaster in these provinces, and heavy rainfall is the most important factor of flash flood disasters. The inter-annual variation in rainfall in these four provinces has a strong impact on the flood-related economic losses in the region, when the investment in flash flood disaster prevention and control projects has little inter-annual variation.

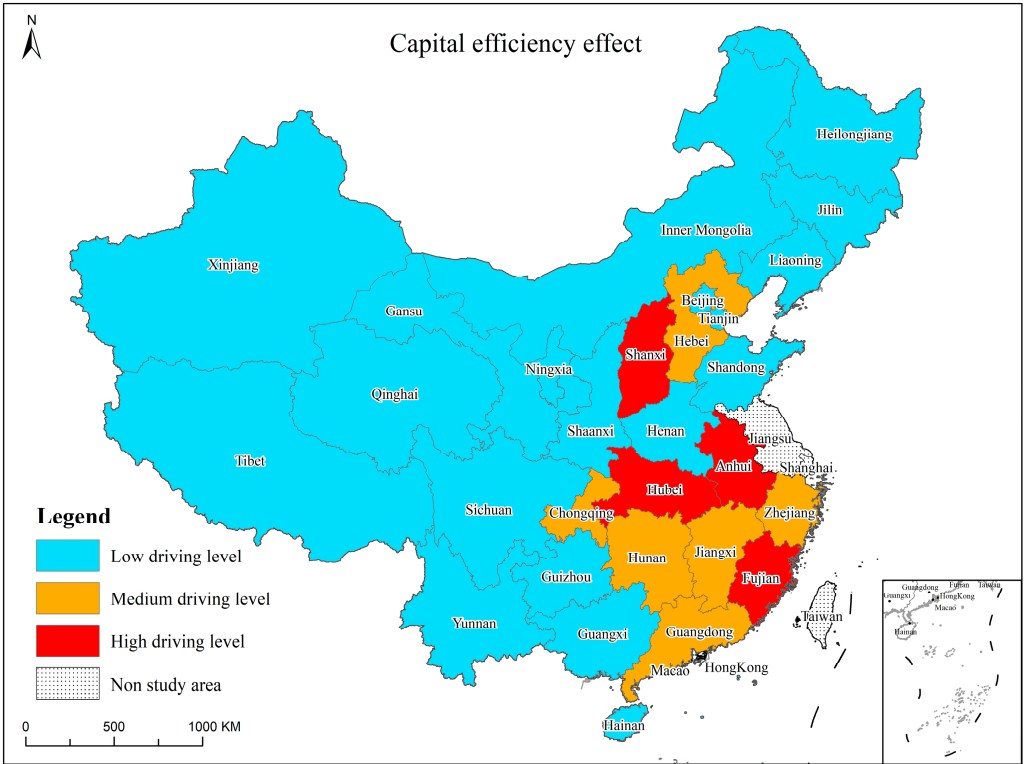

**Figure 7.** Capital efficiency effect driving distribution.

The regions with medium driving levels of capital efficiency effect were Hebei, Zhejiang, Jiangxi, Hunan, Guangdong, and Chongqing (Figure 7). From 2010 to 2020, the average annual rainfall in Hebei, Zhejiang, Jiangxi, Hunan, Guangdong, and Chongqing was 513 mm, 1795 mm, 1861 mm, 1782 mm, 1874 mm, and 1164 mm, respectively. These provinces are also regions where flash flood disasters occur more frequently, and the inter-annual variations in precipitation had a strong impact on the regional flood-related economic loss; however, the effect value was lower than in the high driving level regions.

The regions with a low driving level of capital efficiency effect were Beijing, Tianjin, Inner Mongolia, Liaoning, Jilin, Heilongjiang, Shandong, Henan, Guangxi, Hainan, Sichuan, Guizhou, Yunnan, Tibet, Shaanxi, Gansu, Qinghai, Ningxia, and Xinjiang (Figure 7). Hainan, Guizhou, Yunnan, Sichuan, and Guangxi are in the south of China with abundant precipitation; however, the capital efficiency effect value of these five provinces was obviously smaller than that of strong and medium driving regions, indicating that the inter-annual variation in rainfall had a weak impact on flood-related economic loss. Other provinces are mainly located in the northern part of China, with lower rainfall than the southern region; therefore, the flood-related economic loss in these provinces is relatively low.

### 3.2.5. Loss-Rainfall Effect

The loss-rainfall effect reflects the relationship between flood-related economic losses and annual rainfall in a region. If this effect is characterized by an incremental effect, it indicates that the economic loss of flood disaster in the region presents an increasing trend under the same rainfall intensity. On the contrary, if this effect is characterized by a reduction effect, it indicates that under the same rainfall intensity in the region, the economic loss of flood disaster shows a decreasing trend.

The loss-rainfall effect showed a clear reduction, with an average value of −9.5145 billion Yuan in China, from 2010 to 2020. During the study period, the loss-rainfall effects of most provinces were reduced, indicating that the flood-related economic loss in most provinces and cities is gradually decreasing in terms of unit rainfall intensity.

The provinces with a high driving level of loss-rainfall effect were Shanxi, Anhui, Fujian, and Hubei, and the loss-rainfall effect was incremental (Figure 8). Among these regions, Hebei and Hubei are special. Hebei suffered severe flash floods in 2016, with a single event economic loss of 50.217 billion Yuan. In the same year, the Yangtze River flood disaster occurred in Hubei Province, causing massive economic losses [24]. Although the average loss-rainfall effect of the two provinces was low during the study period, the flood-related economic loss in a single year was extremely serious, leading to the high driving level of the loss-rainfall effect. The province with the highest incremental effect value of loss-rainfall effect was Anhui Province. According to the flood-related economic loss data over the years, the economic loss caused by flood disasters in Anhui Province in 2016 and 2020 reached 500.65 billion Yuan and 600.7 billion Yuan, respectively. The flooding of the Yangtze River in 2016 and the Huai River in July 2020 caused severe economic losses in Anhui Province, leading to a high driving level of loss-rainfall effect in Anhui. The loss-rainfall effect in the other seven provinces showed an obvious reduction. The flood-related economic losses in these provinces under the same rain intensity showed a gradual decreasing trend, indicating that the defense level against flood disasters in these provinces has been greatly improved.

The regions with a medium driving level of loss-rainfall effect included Zhejiang, Guangxi, and Sichuan, and the loss-rainfall effect was incremental (Figure 8). In Zhejiang and Guangxi particularly, typhoons and flash floods are the main types of flood disasters [21,32,33]. Typhoons and flash floods are two disasters that happen very suddenly and without warning. The economic loss caused by typhoons is often especially unpredictable, which is the main reason for the effect value of these two provinces being incremental. Sichuan is in the southwest of China, and the frequent occurrence of regional floods in the province causes great economic losses, which is the main reason for the

incremental effect of the loss-rainfall effect. The absolute value of the effect in these two provinces and Sichuan was lower than that in the high driving effect area.

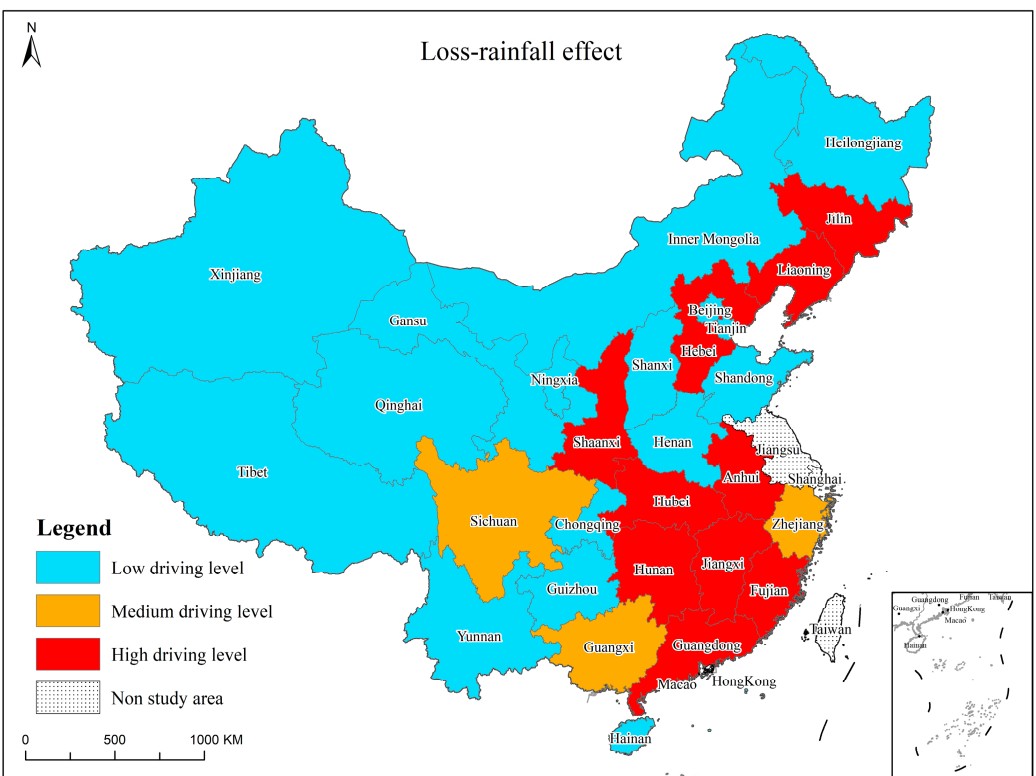

**Figure 8.** Loss-rainfall effect driving distribution.

The regions with a low driving level of loss-rainfall effect include Beijing, Tianjin, Shanxi, Inner Mongolia, Shandong, Henan, Hainan, Chongqing, Guizhou, Yunnan, Tibet, Gansu, Qinghai, Ningxia, and Xinjiang (Figure 8). Among them, in Beijing, Tianjin, Chongqing, Guizhou, Yunnan, and Gansu provinces, the loss-rainfall effect was incremental; moreover, the incremental effect value of Chongqing was higher. However, the flood-related economic losses in these provinces were relatively low in most years; therefore, they are classified as a low driving level. In Shanxi, Inner Mongolia, Shandong, Henan, Hainan, Tibet, Qinghai, Ningxia, and Xinjiang, the loss-rainfall effect showed a reduction. Furthermore, even though the absolute value of the effect was higher in Henan and Hainan provinces, the flood-related economic loss in these two provinces was relatively low during the study period.

## 4. Conclusions

This study analyzed the driving factors of flood-related economic losses in China from 2010 to 2020, using statistics obtained from the National Bureau of Statistics and the Ministry of Water Resources of China. The Kaya identity and LMDI method were used to establish a factor decomposition model to determine what affects economic losses caused by flood disasters. Five effects were isolated, measured, and analyzed: demographic effect, economic effect, flash flood disaster control effect, capital efficiency effect, and loss-rainfall effect.

Using the LMDI method, we deconstructed the driving factors affecting flood-related economic loss; the results show that changes in flood-related economic loss are the result of these five factors. The flash flood disaster control effect showed the most obvious reduction effect. This indicates that the implementation of flash flood disaster prevention and control projects has greatly reduced economic losses caused by flood disasters in China. The continuous development of flash flood disaster prevention and control plays

a positive role in reducing the flood-related economic losses in China. The reduction in the loss-rainfall effect is the inevitable result of the control measures for flash flood disasters. Updating monitoring and warning equipment and the construction of water conservancy informatization have significantly improved the country's flood disaster prevention abilities, effectively reducing flood-related economic losses under the same rainfall intensity.

The demographic effect, economic effect, and capital efficiency effect were shown to be incremental. China is in the later stage of industrialization and the middle stage of urbanization, and the growth of regional economies will also bring greater challenges to flood control work. Regions with higher capital efficiency effects are concentrated in the south of China, where the annual rainfall is generally higher. When the inter-annual variation in flash flood disaster prevention funds is small, the inter-annual variation in precipitation will have a great influence on the change in the effect value. Although China is a low fertility country, the population of most provinces shows an increasing trend, which will inevitably pose more challenges to flood control. Therefore, the demographic effect also shows the characteristics of incremental effect.

Economic development is the foundation of social, scientific, and technological progress, and scientific and technological progress is necessary for the improvement of water conservancy informatization. Considering the GDP statistics of the study period, it is clear that the Chinese economy will maintain a steady growth trend in the future. It also shows that the economic effect in the future will still present the characteristics of the incremental effect. Considering these statistics along with the results of this study, we see that the flash flood disaster control effect shows obvious reduction effect characteristics. Under the condition of future economic growth, the continuous development of the special construction of mountain flood disaster prevention and control will play a positive role in reducing economic loss related to flood disasters.

**Author Contributions:** Conceptualization, Z.Z. and Q.L.; methodology, C.L.; software, Q.M.; validation, L.D.; formal analysis, Z.Z.; investigation, Q.L.; resources, Y.C.; data curation, Y.C.; writing—original draft preparation, Z.Z.; writing—review and editing, Z.Z. and Q.L; visualization, Q.M.; supervision, Q.L.; project administration, L.D.; funding acquisition, Q.L and C.L. All authors have read and agreed to the published version of the manuscript.

**Funding:** The research presented in this paper has been carried out as part of the National Key Research Program (No. 2019YFC1510603), Key Research Program of Guangxi Province (2019AB20003), and Hunan Water Conservancy Science and Technology Project (XSKJ2019081-17).

**Institutional Review Board Statement:** Not applicable.

**Informed Consent Statement:** Not applicable.

**Data Availability Statement:** Not applicable.

**Acknowledgments:** The authors thank the China Institute of Water Resources and Hydropower Research.

**Conflicts of Interest:** The authors declare no conflict of interest.

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
