# Peer review of "Driving Effects and Spatial-Temporal Variations in Economic Losses Due to Flood Disasters in China"

_water, doi:10.3390/w14142266_

Round 1
Reviewer 1 Report
Thank you for the interesting research. Please find my few observations:
1. please cite your figures and table in text. please give references in figure and table captions.
2. please clarify the method and analysis more that others can understand your methodology.
3. please provide contextual explanation for your findings in discussion part.
Author Response
Response to Reviewer Comments
Point 1: please cite your figures and table in text. please give references in figure and table captions.
Response 1: Thank you very much for pointing out the shortcomings of this paper. To solve this problem, We add references to images and tables in relevant places and data sources to the titles of Figures 1 and 4. Other images and tables are calculated for this study, so there are no data reference citations.
Point 2: please clarify the method and analysis more that others can understand your methodology.
Response 2: Thank you for bringing this to our attention. In view of this problem, we added the formula with text explanation to the letters in the formula, so that readers can understand the meaning of each indicator.
Point 3: please provide contextual explanation for your findings in discussion part.
Response 3: Thank you very much for your advice. After full consideration, we add a detailed elaboration in the conclusion section to illustrate my understanding of the research results. Our study points out that as the economy grows, special construction for flash flood control continues to develop, which will have a positive effect on reducing economic losses related to flash floods. Besides, we will further study the relationship between the special construction of flash flood disaster prevention and flood control work.

Reviewer 2 Report
Dear Authors,
I had the chance to read your paper in detail. In general, I find that the content is interesting, however it need some minor improvement. Some comments are:
L73-75: If there is not a specific reason for using parenthesis you can delete it
Section 2.1: I would suggest to put Kaya and LMDI in different sections
Figure 2, 3, 4, 5, 6: Please provide a figure with better resolution
Tables 1 and 2: I assume that the numbers toy are presenting are in Yuan, however this is not clear to the reader
I would really like to see some more insight to the results and how these could help in future planning. You should elaborate more on the management of floods and how your results could help towards having better plans n the future
Author Response
Response to Reviewer Comments
Point 1: L73-75: If there is not a specific reason for using parenthesis you can delete it
Response 1: Thanks for your suggestion. To address this problem, we removed the parentheses that served no purpose.
Point 2: Section 2.1: I would suggest to put Kaya and LMDI in different sections
Response 2: In response to this recommendation, we split the introduction of the LMDI model into a separate section, Section 2.2.
Point 3: Figure 2, 3, 4, 5, 6: Please provide a figure with better resolution
Response 3: In response to this suggestion, we have improved the figures’ resolution.
Point 4: Tables 1 and 2: I assume that the numbers toy are presenting are in Yuan, however this is not clear to the reader
Response 4: Thank you very much for pointing out the shortcomings of this paper. In view of this problem, we add the measurement units of data to the titles of table1 and 2, so that readers can understand.
Point 5: I would really like to see some more insight to the results and how these could help in future planning. You should elaborate more on the management of floods and how your results could help towards having better plans in the future
Response 5: Thank you very much for your advice. After full consideration, we add a detailed elaboration in the conclusion section to illustrate my understanding of the research results. Our study points out that as the economy grows, special construction for flash flood control continues to develop, which will have a positive effect on reducing economic losses related to flash floods. Besides, we will further study the relationship between the special construction of flash flood disaster prevention and flood control work.
